# Modern Rape Myths: Justifying Victim and Perpetrator Blame in Sexual Violence

**DOI:** 10.3390/ijerph20031663

**Published:** 2023-01-17

**Authors:** Carol Murray, Carlos Calderón, Joaquín Bahamondes

**Affiliations:** School of Psychology, Universidad Católica del Norte, Avenida Angamos 0610, Antofagasta 1240000, Chile

**Keywords:** modern rape myths, gender system justification, attribution of blame, mediation

## Abstract

Rape myths are beliefs, stereotypes, and attitudes usually false, widespread, and persistent about rape, victims, and perpetrators. They aim to deny and justify men’s sexual assault against women. This study evaluates the mediating effect of modern rape myths on the relationship between gender system justification and attribution of blame to both victim and perpetrator in a fictional case of sexual violence. A total of 375 individuals residing in Chile, 255 women and 120 men, 19–81 years (*M* = 37.6 *SD* = 13.06) participated in the study. Results from a Structural Equation Model show that gender system justification is directly related to the attribution of blame to the victim, showing an indirect relationship throughout the modern rape myth. However, gender system justification and attribution of blame to the aggressor are indirectly related, being mediated by modern rape myths. The study of the relationship between the acceptance of modern rape myths, gender-specific system justification, and victim and aggressor blame for rape is a contribution to understanding beliefs justifying sexual violence against women.

## 1. Introduction

On 6 November 2018 in Ireland, a man accused of raping a 17-year-old girl was acquitted [1]. It was deemed that the woman was seeking sexual interaction since she was wearing a “sexy” gli slip, interpreted as a sign of consent [1]. In 2017 in Italy, two rape convicts were acquitted. The court posed that the woman was very masculine, making it unlikely that she had been raped [2]. Similarly, “la manada” in Spain and Antonia Barra in Chile are cases questioning what is meant by sexual consent, promoting discussion about sexual violence beliefs [3,4].

The victim’s clothing, lifestyle, and setting where sexual aggressions occur are often data used to evaluate cases of sexual violence and attribute responsibility to the victim and/or blame the aggressor [2]. These attributions are based on beliefs that attempt to justify, minimize or deny the existence of sexual aggressions against women, trivializing and naturalizing the problem of gender violence [4]. According to Pan American Health Organization [5], all countries report alarming numbers of sexual crimes, which makes sexual violence a worldwide problem mainly affecting women, one of the most vulnerable groups in society [6]. Studies worldwide report up to 58.8% prevalence. These figures reflect the transversality of the problem, affecting women from different social conditions [7,8,9].

The consequences experienced by the victims are extensively documented, showing the need to generate knowledge to address this problem efficiently and effectively [10]. This has made sexual violence gain relevance in recent decades, becoming an activism issue for different groups and social movements [11]. Feminist movements, such as #MeToo, question hegemonic and dominant discourses promoting victim and perpetrator stereotypes denying, biasing, and justifying sexual and gender-based violence [11,12,13]. These movements demand changes in social structure at all levels to guarantee the effective decrease of sexual violence against women and other vulnerable groups [14,15].

One of the conceptualizations intending to explain how sexual violence is legitimized and validated is rape myths [16]. Rape myths are defined as culturally ingrained, false, widespread, and persistent attitudes and beliefs to justify, deny or normalize men’s sexual aggressions against women [13,16,17,18]. These beliefs produce bias in the attributions toward those involved in sexual aggression, making the victim responsible and reducing the aggressor’s blame [19,20,21].

The first empirical approach to this construct was developed by Burt [16], who proposed a measurement scale using direct and explicit language to refer to rape-related beliefs, called *Rape Myth Acceptance Scale* (RMA). Similarly, Payne et al., [13] developed *Illinois Rape Myth Acceptance Scale* (IRMA) to assess rape myth adherence. This scale maintains classic and direct language in measuring these myths [13]. Gerger et al., [17] following studies about modern racism and sexism, propose modernizing language to assess rape myths. They developed an instrument with less obvious statements about rape and other forms of sexual assault, using subtle, indirect, and implicit language [17,22].

Research on rape myths in their classical expression is conducted mainly in the USA [23]. On the other hand, evidence regarding modern rape myths is mainly obtained in Europe, with few studies in Latin America [22,24]. In Chile, the modern approach to rape myths is investigated from a criminological perspective [25]. Therefore, providing evidence on the relationship between modern rape myths and ideological variables -which are related in other studies - is novel and necessary in a Latin American context [23].

The study of classical and modern rape myths provides evidence to understand how these myths operate in individuals and groups [23]. These findings show that rape myths are closely related to beliefs that validate hierarchical order among individuals and groups. Beliefs that naturalize social order and control, maintaining status quo, are characterized by highly conservative thinking. One of the most studied beliefs is *system justification* related to rape myths and responsibility attributed to the victim [26,27]. System justification theory proposes that people maintain basic motivations that reinforce the belief that the established social order is necessary and convenient [28,29]. This theory poses that stereotypes characterizing social categories allow rationalizing and reinforcing the specific characteristics of intergroup relationships. In gender stereotype cases, they would allow supporting and tolerating inequalities based on gender relations [30].

*System justification associated with gender differences* (GSJ), i.e., the belief that relationships between men and women are fair and legitimate, aims to minimize and/or naturalize gender injustice; however, it has positive effects on women’s subjective psychological well-being [31]. Evidence suggests that women who show high GSJ are more likely to rationalize gender discrimination, even attributing responsibility for the discrimination experienced, supporting the belief that it is deserved and fair [32].

In this regard, previous studies found a significant relationship between GSJ, rape myths, and victim attributions of responsibility. Particularly, Martini and De Piccoli [33] show that two classic rape myth dimensions (“she asked for it” and “it wasn’t really rape”) explain the association between GSJ and an observer’s lack of motivation to intervene in a fictional sexual violence situation. These results show that high GSJ levels would be related to high levels of adherence to rape myths, which, in turn, would be associated with low intention to intervene in sexual assault situations [33].

In brief, people who adhere to rape myths deny, minimize, and trivialize the problem of sexual violence [26]. Such denial is associated with the legitimization of women’s subordination to men, justifying sexual aggressions [33]. Invisibilizing sexual violence preserves the male domination system. Therefore, blaming the aggressor implies assuming that social order is unjust [34]. However, blaming the victim maintains the belief that the system is fair to those who conform to cultural stereotypes [34].

### The Study

As mentioned above, previous evidence shows a mediating effect of rape myths on the relationship between gender system justification and an observer’s intention to intervene in a fictional sexual violence situation [33]. Martini and De Piccoli [33] in their study use the measure of classic rape myths as a mediating variable. Here the aim is to assess the mediating effect of a measure of modern rape myths. Additionally, a measure of blame attributed to both victim and perpetrator in a fictitious case of sexual violence as the dependent variable is used here. The hypothesized model is shown in Figure 1.

Results are expected to show a full mediation effect of modern rape myths on the relationship between gender system justification and attribution of blame to both victim and perpetrator. Particularly, high levels of gender system justification should be related to higher victim-attributed blame and lower offender-attributed blame. Additionally, in terms of indirect effects, higher levels of gender system justification should be positively related to the adherence to modern rape myths, while these should be positively related to victim-attributed blame and negatively related to perpetrator-attributed blame.

## 2. Methodology

### 2.1. Design and Type of Study

A correlational study was conducted, as the relationships between variables are explored through a cross-sectional design according to research design classification in psychology [35].

### 2.2. Participants

The sample consists of N = 375 subjects residing in Chile, 255 women and 120 men. Ages range from 19 to 81 years (*M* = 37.6 *SD* = 13.06). Table 1 shows the results of the main demographic variables.

Regarding religious beliefs, 41.3% do not identify with any religion, followed by 33.9% who identify with the Catholic religion; 8.5% are agnostic, 6.1% evangelicals, and 10.2% identify with other religions. In terms of religiosity, 25.3% say they are not religious, 26.4% identify themselves as not very religious, 36.3% are moderately religious and 12% are very religious.

The socioeconomic characteristics, most of them report that the person who provides the main family income works as a manager or independent professional in traditional careers (law, medicine, engineering, etc.), representing 47.7% of the sample.

A total of 38.5% participants reports that the person who contributes with the main income to the family holds a professional title, followed by 22.7% with a graduate degree, and 0.3% with no formal education.

### 2.3. Procedure

The survey was administered online using the Qualtrics platform. Data collected corresponds to surveys completed between August and December 2021.

The sampling was non-probability purposive and used snowball technique to reach a larger number of people. Participants were contacted through e-mails and messages on different social media platforms (e.g., Facebook, WhatsApp, Twitter). Initial contact was made through a brief presentation of the study, in which researchers’ data and general aspects necessary to participate were provided (e.g., residing in Chile, being over 18). The homepage of the survey included an informed consent form, which provided further data on the purpose and implications of participating.

### 2.4. Instruments

#### 2.4.1. Sociodemographic Questionnaire

A sociodemographic characterization questionnaire was used, consisting of 10 questions that included variables such as sex, gender identity, sexual orientation, age, educational level, and political tendency, among others.

#### 2.4.2. Modern Rape Myths

This construct was measured with the acceptance of modern myths about sexual aggression scale [17], which has a validated version for its administration on Chilean population [22]. This instrument addresses rape myths related to sexual violence denial, antagonism towards victims’ demands, lack of support for policies to help victims, beliefs normalizing male sexual coercion, and beliefs blaming the victim and excusing the aggressor. For example: *“When a single woman asks a single man to come over, she is indicating that she is not reluctant to have sex”* [22]. The Chilean instrument consists of 14 items and 6 response alternatives ranging from “totally agree” to “totally disagree”. Reliability is α = 0.92, a value close to the results obtained from the adaptation and validation study on Chilean population. Data show a good fit to the unidimensional model [22].

#### 2.4.3. Gender-Specific System Justification

It was measured with six items from gender-specific system justification scale [30]. Gender-specific system justification measures the extent to which people believe that gender relations are fair in the cultural and social context they live in, e.g., *“In general, relationships between men and women are fair”*. The one-dimensional scale consists of 8 items, responses being categorized using a 6-point Likert scale, ranging from “strongly disagree” to “strongly agree” [30]. Six items of the scale were used because the two others showed factorial saturations lower than 0.250. Sample reliability is α = 0.83.

#### 2.4.4. Blame Attributed to a Man or Woman in a Rape Situation

It was measured with items from Sexual Assault Culpability Measures (CFA) scales proposed by Persson and Dhingra [36]. The first one assesses blame attributed to a woman and the second one to a man, in a fictional account of a woman’s rape. The items in each scale are the same, changing only the word “woman” for “man”, depending on whom blame is attributed to, e.g., *“This incident was due to the woman/man”*. Each item is scored on a 6-point Likert-type scale from “strongly disagree” to ”strongly agree”. The higher the score, the greater the attribution of blame to the victim or aggressor. Eight items were used to measure both, blame attributed to the victim and to the offender. Reliability for attribution of blame to the victim is α = 0.85 and α = 0.93 for the aggressor, both being satisfactory reliability indices [37].

## 3. Results

### 3.1. Bivariate Correlations

Table 2 shows the results of Pearson’s r bivariate correlations between the variables used in the study. All scales are significantly related (*p* < 0.05). Modern rape myths are positively related to GSJ, while attribution of blame to the victim is the one showing the highest value in the correlation, and negatively related to attribution of blame to the aggressor. On the other hand, GSJ relates positively to attribution of blame to the victim and negatively with attribution of blame to the aggressor, the latter being the least intense correlation.

### 3.2. Mediation Model

To evaluate the hypothesis that acceptance of modern rape myths mediates the relationship between gender system justification and attribution of blame to victim and perpetrator, Structural Equation Model (SEM) was fitted using Mplus 7.4 [38]. The fitted model corresponds to the hypothesized model in Figure 1. The weighted least square mean and variance adjusted (WLSMV) estimator was used, a method recommended for categorical variables [38]. The model analyzing the mediation of modern rape myths between GSJ and victim and perpetrator blame shows good fit indices (Table 3), RMSEA = 0.058, CFI = 0.975, TLI = 0.973.

Figure 2 shows the standardized values of the model, where gender system justification is significantly related to modern rape myths (*B* = 0.72, 95% CI 0.68, 0.78, *p* < 0.001). Modern rape myths are positively related to victim blaming (*B* = 0.71, 95% CI 0.63, 0.80, *p* < 0.001) and negatively to aggressor blaming (*B* = −0.57, 95% CI −0.67, −0.47, *p* < 0.001). On the other hand, GSJ is positively and significantly associated with attribution of blame to the victim (*B* = 0.13, 95% CI 0.04, 0.23, *p* < 0.001). The relationship between GSJ and attribution of blame to the aggressor is non-significant (*B* = −0.07, 95% CI −0.18, 0.04, *p* = 0.199). Consistent with the hypotheses, the indirect effect between GSJ and attribution of blame to the victim is *B*${}_{indirect}$ = 0.52, 95% CI 0.45, 0.58, *p* < 0. 001, while the indirect effect of GSJ and attribution of blame to the offender is *B*${}_{indirect}$ = −0.42, 95% CI −0.50, −0.34, *p* < 0.001.

Results suggest that modern rape myths mediate the relationship between gender system justification and attribution of blame to the victim and perpetrator. Subjects showing higher gender system justification also show high adherence to rape myths, tending to attribute more blame to the victim in a fictional rape scenario. However, in the case of attribution of blame to the aggressor, the relationship is inverse, i.e., the greater the justification of the gender system and acceptance of modern myths, the lower the attribution of blame to the aggressor, being modern rape myths the way in which they justify sexual aggression. In other words, modern myths are one of the mechanisms by which male behavior and rape are justified. In examining the results of the relationship between JSG and attribution of blame to the aggressor, the correlation is moderate but significant. However, it masks the variable of modern rape myths as a mediator of the relationship [39].

## 4. Discussion

This study aimed to evaluate the mediating effect of modern rape myths between gender system justification and the attributions of blame to the victim and aggressor in Chilean population. We hypotesis that modern rape myths would completely mediate the relationship between gender system justification and the attributions of blame to the victim and aggressor. A structural equation model was used to test our prediction, showing good overall fit indices in the Chilean sample [37].

Results show that modern rape myths mediate the relationship between gender system justification and attribution of blame to the victim and the perpetrator. However, a novel finding is the difference between the type of mediation (full or partial) of modern rape myths between gender system justification and victim and perpetrator blaming. Results partially support the hypothesis that modern myths would fully mediate the relationship between GSJ and blame attribution to victim and perpetrator since, according to the results, only the relationship with the attribution of blame to the perpetrator supports this hypothesis. The last finding is similar to the results found by Martini and De Piccoli [33], who report that classical rape myths fully mediate the relationship between GSJ and the bystander’s intention to intervene in a fictional rape situation.

In the case of victim blaming are directly related. Victim blaming increases when people believe that gender relations are fair and adhere to myths proposing that sexual violence only occurs in very specific contexts and to a certain “type of woman” [12,13,16]. Both beliefs are positively related to the attribution of blame to the victim in a woman’s rape situation [18,19,20].

As to the relationship between gender system justification and the attribution of blame to the aggressor, people minimize and justify the aggressor’s actions through modern rape myths when they believe that the woman transgresses social norms via improper and inadequate behavior [13,18], diminishing “man’s responsibility” by normalizing and legitimizing sexual aggressions committed against women [20].

Overall, gender system justification is strongly associated with modern rape myths, showing that they are a way of justifying the status quo of gender asymmetries [33]. In countries with highly patriarchal cultures such as Chile, machismo naturalizes men’s domination over women, promoting unequal gender relations [40,41]. Consistent with our findings, believing in the fairness and legitimacy of gender-based relations in Chile is closely tied to endorsing rape myths—that is, to justify the gender status quo entails accepting that women are mostly to blame for experiencing rape, and that male perpetrators are just acting on expected, natural urges. These attitudes exonerate men from their responsibilities, thus protecting their privileged position in society [30,33]. In this way, women’s subordination is justified as a measure and necessity to maintain the established social order [42].

GSJ, as well as modern rape myths, minimize, deny, and justify sexual violence against women, promoting and reinforcing beliefs that validate women’s responsibility for sexual assault [33,43]. Indeed, attributing blame to the victim upholds the idea that the world is fair and that “everyone gets what he/she deserves”, promoting the belief of a universal justice that legitimized intergroup inequalities [44], installing discourses that promote conservative gender relationship stereotypes [1,12]. They also diminish the aggressor’s blame with discourses underestimating the problem of sexual violence and those normalizing men’s sexual aggression behaviors, by viewing men as having a natural and instinctive sexuality that is difficult to control rationally, and that should not be provoked by women [13].

This agrees with the theory posing that epistemic needs are satisfied by justifying the system, which would explain why men and women tend to attribute greater blame to female rape victims and acquit male aggressors, as oppressors and oppressed, respectively, along with weakening identity and perceiving one’s own social disadvantage [31,32].

In summary, GSJ is directly related to the attribution of blame to the victim, also showing an indirect relationship throughout the modern rape myth. As to GSJ and the attribution of blame to the aggressor, this relationship is indirect and explained by modern rape myths. In other words, someone who justifies women’s inequalities and subordination may easily attribute blame to the victim directly in a sexual aggression situation. However, modern rape myths are used to diminish the victimizer’s blame.

This study has some limitations important to consider for analyzing results. The first relates to the sample since a significant number of respondents reports belonging to families with higher educational and economic levels, a sector of the population that is not a majority in Chile. This can generate a participation bias that affects the representativeness of the results [45]. This bias may be associated with the online application of the survey; however, evidence shows that there are no differences in the response rate of these variables between the online and paper application [46]. So, it is important that future studies include respondents from lower educational and economic levels.

The second limitation is the low participation of men in the survey, representing only one-third of the total sample, a larger group may compromise the generalizability of results.

Another limitation is related to the design. Since the study is cross-sectional, it is not possible to refer to causality in the relationships between variables. Given this general belief, it is reasonable to conceptualize GSJ as an antecedent, considering that the effects observed are consistent with its role as a distal variable in the model. The causal relationship of the GSJ is observed in other contexts [47,48], so experimental or longitudinal studies are necessary to establish this type of relationship with the variables studied.

## 5. Conclusions

In conclusion, results support the hypothesis that modern rape myths are a way of justifying gender inequalities, which seek to maintain the subordination of women to men. The attribution of blame to the victim serves as a punishment since it is believed that women provoke and expose themselves to the occurrence of the crime by transgressing the mandate and social order of subordination. Therefore, it is a disciplinary measure to hold them responsible for sexual aggression [42].

## Figures and Tables

**Figure 1 ijerph-20-01663-f001:**
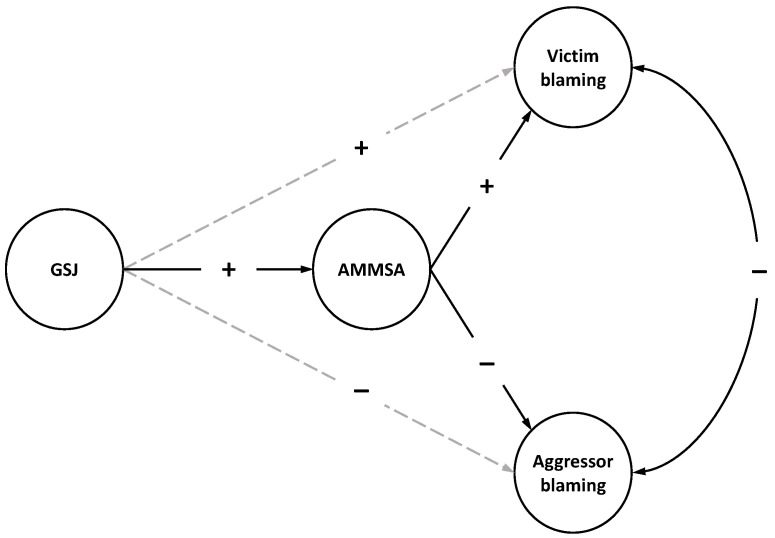
Hypothesized model of rape myth justification.

**Figure 2 ijerph-20-01663-f002:**
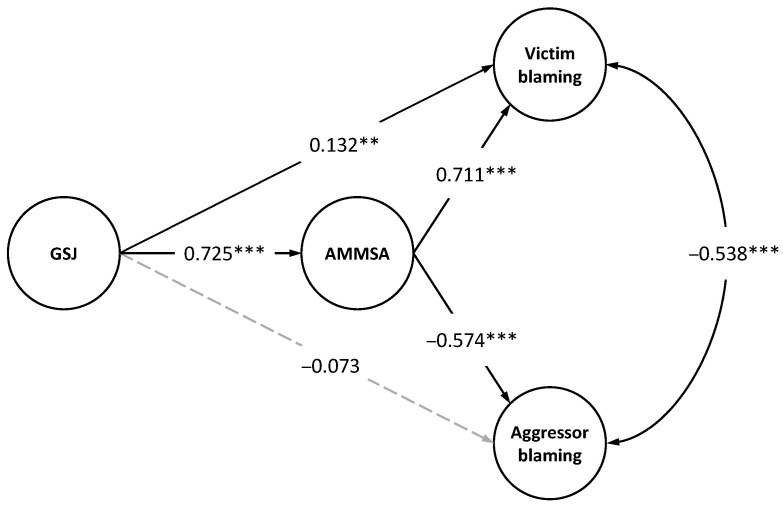
GSJ and attribution of blame to the victim and the aggressor, mediated by modern rape myths; Standardized values, ** < 0.01, *** < 0.001.

**Table 1 ijerph-20-01663-t001:** Sociodemografic characteristics of the sample.

Variable	Category	*N*	%
Sex	Man	120	32
	Women	255	68
	Total	375	100
Gender Identity	Male	115	30.7
	Female	249	66.4
	Other	11	2.9
	Total	375	100
Sexual Orientation	Heterosexual	323	86.1
	Homosexual	19	5.1
	Other	33	8.8
	Total	375	100
Marital Status	Married or cohabiting	129	34.4
	Separated or divorced	29	7.7
	Single	215	57.3
	Widowed	2	.5
	Total	375	100
Political Orientation	Right	31	8.3
	Center-Right	41	10.9
	Center	82	21.9
	Center-left	83	22.1
	Left	138	36.8
	Total	375	100

**Table 2 ijerph-20-01663-t002:** Bivariate correlation, mean, and standard deviation.

Variable	1	2	3	M	SD
Modern Rape Myth (1)	-			30.03	13.77
Gender System Justification (2)	0.580 **	-		14.35	6.07
Victim Blaming (3)	0.719 **	0.514 **	-	13.58	5.91
Aggressor Blaming (4)	−0.437 **	−0.283 **	−0.513 **	43.44	6.99

** < 0.01.

**Table 3 ijerph-20-01663-t003:** Model fit indices.

	χ${}^{2}$ (df)	RMSEA	95% CI	IFC	TLI	*p*
Model	1323.39(588)	0.058	[0.054–0.062]	0.975	0.973	<0.001

## Data Availability

The data is private and can only be reviewed by the researchers and the members of the ethics committee of the Universidad Católica del Norte.

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
