# Peer review of "Modern Rape Myths: Justifying Victim and Perpetrator Blame in Sexual Violence"

_ijerph, 2023, doi:10.3390/ijerph20031663_

Round 1
Reviewer 1 Report
Review of IJERPH-2098613
Modern rape myths: Justifying victim and perpetrator blame in sexual violence.
December 16, 2022
This study provides evidence for relationships among acceptance of modern rape myths, gender-specific system justification, and victim and aggressor blame for rape in a Chilean sample. The literature review is sufficient, the method sound, the results clearly stated, limitations address, and conclusions reasonable. I have only a few minor comments, given below.
1. What are some examples of the “different social media platforms” used, and could the choices of platforms (or even the online context of the survey) led to the over-sampling of higher socioeconomic participants?
2. Could the authors provide a full listing of the 10 sociodemographic categories they used?
3. The sentence on p6, “In other words, modern myths are the mechanism by which male behavior and rape are justified” is over-stated. Certainly they are not the only such mechanism?
4. On p8, while it may be “reasonable to conceptualize GSJ as an antecedent,” this is an extrapolation from the present results and should be characterized as such.
Author Response
Dear reviewer,
attached is a response in pdf format

Reviewer 2 Report
Dear authors, this article is interesting on sexual violence issues. I suggest elaborating on some points in the discussion about the mediating effect of a measure of modern rape myths. In addition, the conclusion could be separated from the discussion section, so it will be clearer and more powerful to recommend future research.
Author Response
Dear reviewer
attached is a response in pdf format

Reviewer 3 Report
I would recommend to provide the English version of references 5, 6, 7.
Author Response

(The authors gave the same response as above.)
